

# Effect of prolactin on cytotoxicity and oxidative stress in ovine ovarian granulosa cells

Ruochen Yang[1,*], Shuo Zhang[2,*], Chunhui Duan[1], Yunxia Guo[1], Xinyu Shan[1], Xinyan Zhang[1], Sicong Yue[1], Yingjie Zhang[1] and Yueqin Liu[1]

[1] Hebei Agricultural University, Baoding, China
[2] China Agricultural University, Beijing, China
[*] These authors contributed equally to this work.

## ABSTRACT

**Background.** Prolactin (PRL) has been reported to be associated with oxidative stress, which is an important contributor leading to cell apoptosis. However, little is known about the mechanisms underlying the effects of PRL on cytotoxicity and oxidative stress in ovine ovarian granulosa cells (GCs).

**Methods.** Ovine ovarian GCs were treated with 0, 4, 20, 100 and 500 ng/mL of PRL. Then, the cytotoxicity, cell viability, malondialdehyde (MDA), reactive oxygen species (ROS), superoxide dismutase (SOD) and total antioxidant capacity (T-AOC) of GCs were detected. Additionally, 500 ng/mL PRL was chosen as the high PRL concentration (HPC) due to its high cytotoxicity and oxidative stress. Proteomic and metabonomic were performed to examine the overall difference in proteins and metabolic pathways between C (control: 0 ng/mL PRL) and P groups (500 ng/mL PRL).

**Results.** The results indicated that GCs treated with 4 ng/mL PRL significantly decreased ($P < 0.05$) the cytotoxicity, ROS and MDA, increased ($P < 0.05$) the cell viability, SOD and T-AOC, and the GCs treated with 500 ng/mL PRL showed the opposite trend ($P < 0.05$). Supplementation with 500 ng/mL PRL significantly increased the proteins of MT-ND1, MAPK12, UBA52 and BCL2L1, which were enriched in ROS and mitophagy pathways. Pathway enrichment analysis showed that the pentose phosphate pathway was significantly enriched in the P group.

**Conclusion.** A low concentration of PRL inhibited cytotoxicity and oxidative stress. HPC induced oxidative stress in ovine ovarian GCs via the pentose phosphate pathway by modulating the associated proteins MT-ND1 in ROS pathway and UBA52, MAPK12 and BCL2L1 in mitophagy pathway, resulting in cytotoxicity.

## INTRODUCTION

Oxidative stress is defined as an excessive level of intracellular reactive oxygen species (ROS) production due to the imbalance between oxidation and antioxidant. ROSs includes superoxide anion ($O^{2-}$), hydrogen peroxide ($H_2O_2$), and hydroxyl radicals (-OH.), which can be scavenged by antioxidants (*Schieber & Chandel, 2014*). It has been proposed that oxidative stress can impede the reproduction function of the body (*Meniri et al.,*

Corresponding authors
Yingjie Zhang,
zhangyingjie66@126.com
Yueqin Liu, liuyueqin66@126.com

*2022*). A study by *Stier et al. (2012)* indicated that pre-reproductive oxidative damage was significantly related to mice's litter size at birth. The oxidative stress generated at high altitudes or under low air pressure can affect the development and function of the sheep corpus luteum, further leading to reduced fertility (*Parraguez et al., 2013*). The ROS and $O^{2-}$ radicals may be involved in human reproduction (*Drejza et al., 2022*; *Veena et al., 2008*), while oxidative stress is the most frequent cause of female infertility disorders, including polycystic ovary syndrome (PCOS) (*Mohammadi, 2019*). Additionally, previous studies have demonstrated the critical physiological role of ROS, which was locally produced by endothelial cells, neutrophils and macrophages within the follicle during folliculogenesis and ovulation (*Mitchell & Johnston, 2022*; *Abedelahi et al., 2010*; *Hennet, Yu & Combelles, 2013*). *In vitro*, plenty of evidence showed that oxidative stress is responsible for the abnormal growth and function of granulosa cells (GCs) during ovarian follicular development (*Zhang et al., 2016a*; *Zhang et al., 2016b*; *Li et al., 2016*). Subsequently, GCs apoptosis leads to follicular atresia (*Yuan et al., 2016*) and, more seriously, oocyte and ovarian dysfunction (*Lai et al., 2018*).

Prolactin (PRL) is a protein hormone synthesized and secreted by several cells and tissues in the body, such as anterior pituitary, mammary glands, T-lymphocytes and hypothalamus (*Henriques et al., 2022*). The normal serum PRL levels in females, depending on reference values, range from nearly 2 to 25 ng/mL (average 13 ng/ml) (*Melmed et al., 2011*). But the chemical immunoassay showed that the serum PRL levels of females (Table 1) were different in different periods (*Lu et al., 1983*). PRL in follicular fluid was always found to be at a higher concentration than in serum (*Kamel et al., 1994*). A previous study has shown that the normal content of PRL in small follicles (<eight mm) is 35.42 ± 3.63 ng/ml, significantly higher than that in large follicles with 24.53 ± 2.50 ng/ml PRL (*Gu et al., 1993*). The normal serum concentrations of PRL in ewes ranged between 12 and 24 ng/mL (17.8 ± 1.5 ng/mL, on average) (*Caja et al., 2020*). A previous study has shown that 10 ng/mL PRL can promote endothelial cell proliferation and capillary formation (*Malaguarnera et al., 2002*), whereas 2 μM PRL dramatically increased the NK-mediated killing of the K562 cell line (*Mavoungou, Bouyou-Akotet & Kremsner, 2005*), indicating that low concentrations of PRL promote cell proliferation, but high concentrations increase cytotoxicity. Some studies also emphasized that different doses of PRL can alter oxidative-antioxidant balance in mammals (*Farmer, Lapointe & Cormier, 2017*; *Rodak et al., 2022*). *Thebanlt (2017)* showed that PRL helps the retinal pigment epithelium to survive *via* antioxidant actions, while *Rodak et al. (2022)* found that treatment with 300 ng/mL of PRL induced oxidative stress in human MIN6 cells. PRL excess (hyperprolactinemia) may lead to hypogonadism, which can aggravate the oxidative stress in the body (*Veena et al., 2008*). Meanwhile, a previous study has reported that higher PRL concentration can generate reproductive disorders by inducing oxidative stress and damage (*Veena et al., 2008*; *Hilali et al., 2013*). At the same time, females with endocrine abnormalities also have high PRL levels in the blood, creating hyperprolactinemia, which causes insufficient ovarian function, overflowing milk, and in severe cases, affecting follicle growth (*Chen, Fu & Huang, 2016*). Therefore, the influence of high PRL concentrations on cellular oxidative stress and physiological functions require more attention.

**Table 1  The serum PRL levels of females was different in different periods.**

| Item | Periods | | | | |
|---|---|---|---|---|---|
| | Follicular phase | Luteal phase | Pregnancy (1–3 months) | Pregnancy (4–6 months) | Pregnancy (7–9 months) |
| PRL (ng/mL) | 17.30 ± 2.31 | 21.18 ± 3.29 | 47 ± 9.1 | 104 ± 2.2 | 348 ± 33.3 |

In this study, we investigated the impact of varied concentrations of PRL on GCs cytotoxicity and oxidative stress, intending to establish a theoretical foundation for enhancing GCs growth and addressing functional irregularities. It will provide a basis for future applications of PRL in ruminant reproduction.

## MATERIAL AND METHODS

### Cell collection

Abnormal follicle development can lead to decreased fecundity of sheep and affect production (*Zhang et al., 2016a*; *Zhang et al., 2016b*; *Li et al., 2016*). At the same time, it is also believed that the dysfunction of granulosa membrane cells caused by high concentration of PRL reduces the production of steroid hormones in ovary, which affects the development and maturation of follicles, leading to abnormal menstruation and ovulation disorders and infertility. Therefore, we choose sheep as the model to lay the foundation for the future application of PRL in ruminant reproduction and to provide a theoretical basis for the impact of PRL on human reproduction.

Fresh ovaries from twelve sexually mature ewes aged between 1 and 1.5 years were collected at the local cooperative abattoir (Baoding, Hebei, China), kept at 37 °C, then transported to the laboratory in a buffered saline solution supplemented with streptomycin/penicillin mixture (1%). Dominant follicles with 3–7 mm diameter were isolated from the ovaries after 3 times sterile Dulbecco's Phosphate Buffered Saline (DPBS) cleaning. Then GCs were collected from the interior of the follicles with a one mL injector and filtered with a 100 mm filter. The filtrated liquid was centrifuged at 1,500 rpm for 10 min to obtain the precipitate. Then the precipitate was lysed in one mL sterile red blood cell (RBC) lysate for 3 min to dissolve the RBC. Harvested cells were washed with DPBS ≥ 3 times and used for *in vitro* culture. The growth of GCs was observed under a microscope, and non-adherent cells were removed. All procedures used in this study were approved by the Laboratory Animal Ethics Committee of Hebei Agricultural University (Hebei, P.R. China; permit number 2023045).

### Cytotoxicity and cell viability assay

The GCs were seeded in 6-well-culture plates at a density of $6 \times 10^4$ per well and treated with 0, 4, 20, 100 and 500 ng/mL PRL (PROSPEC, cyt-240, purity ≥99.0%) for 24 h to validate the toxicity of PRL. The morphology of GCs was observed by inverted phase contrast microscope to evaluate the cytotoxicity with different concentrations of PRL (normal ovarian GCs are usually spindle-shaped, while the shape changes when the external environment changes).

The direct cytotoxicity of materials was evaluated as percentage cell viability (*Troiano et al., 2018*) using the Cell Counting Kit 8 assay (CCK-8, ck-04; DOJINDO Laboratories, Munich, Germany). GCs were seeded in 96-well-culture plates at a density of $5 \times 10^3$ per well and treated with 0, 4, 20, 100 and 500 ng/mL PRL for 24 h, respectively. Then a 10 μL CCK-8 was added into each well, and the cells were hatched in the dark for 1-4 h. A microplate reader was used to quantify the optical density of the cells at 450 nm. Cell viability values were calculated based on test data. Cell survival rate $\gamma$ calculation common formula is $\gamma = [(As\text{-}Ab)/(Ac\text{-}Ab)] \times 100\%$ (As: Optical density of P group, Ac: Optical density of C group, Ab: Optical density of the blank well).

## Oxidative stress index detection
### Oxidative stress assay
GCs incubated with 0, 4, 20, 100 and 500 ng/mL PRL were cultured in a 6-well plate at a density of $6 \times 10^4$ per well and then trypsinized and collected in 1.5 ml centrifuge tubes. After incubation, cells were immediately centrifuged twice at $1000 \times$ g for 10 min to remove the supernatant to obtain GCs, which were kept at $-80\,°C$ prior to analysis.

Superoxide dismutase (SOD) activity (A001-3; Nanjing Jiancheng Bioengineering Institute, Jiangsu, China), malondialdehyde (MDA, Nanjing Jiancheng Bioengineering Institute, A003-1) and total antioxidant capacity (T-AOC) (A015-3-1, Nanjing Jiancheng Bioengineering Institute, Jiangsu, China) were measured following the manufacturer's instructions with respective analytical kits.

### ROS detection
GCs incubated with 0, 4, 20, 100 and 500 ng/mL PRL were cultured in a 96-well plate at a density of $5 \times 10^3$ per well and then incubated with fluorescent probe of ROS (H2DCF-DA, Ex/Em = 488/525 nm) for 20 min in dark at 37 °C. Then, incubated cells were washed twice with DMEM/F12 and images were captured under an inverted fluorescence microscope (Leica DM IRB, Leica, Wetzlar, Germany) using a green-fluorescence filter and images were analyzed using ImageJ 1.48v (National Institutes of Health, Bethesda, MD, USA).

## Experimental design
### The GCs model of PRL
In the present study, we found that cells treated with 500 ng/ml PRL showed significantly lower cell viability and higher cytotoxicity and oxidative stress than other groups. By taking into account the results of prior investigations (*Neradugomma, Subramaniam & Tawfik, 2014*), the concentration of 500 ng/mL PRL was selected as the model for the subsequenteffects of high concentrations of prolactin (HPC) on cytotoxicity and oxidative stress in ovarian GCs. Moreover, a serum PRL level greater than 500 ng/mL is diagnostic of a macroprolactinoma (*Vilar et al., 2008*). Then, the GCs were treated with 0 ng/mL (C group) and 500 ng/mL ovine PRL (P group), with 6 replicates in each group. The mechanism of HPC regulating oxidative stress in ovine ovarian GCs was studied using integrated proteomic and metabolomic approaches.

*Sample collection*

GCs in 6-well plates from C and P groups were collected according to previously described methods (*Yao, Xiao & Zhou, 2021*).

## Proteomic analysis
### Protein extraction

Samples were shredded with liquid nitrogen, lysed in 8 M urea lysis solution, and then stored on ice for 1 h, followed by sonication for 5 min. The lysate was centrifuged for one hour at 12,000 × g and 4 °C. The supernatant was transferred to a clean tube. The Bradford protein test was used to determine protein concentration. Each sample's extracts were reduced with 5 mM DTT at 56 °C for 25 min before being alkylated by supplementation of 14 mM Iodoacetamide at room temperature for 30 min in the dark. The urea concentration from each sample, containing 2 M at less, was digested with Tyrisin at 1:100 enzyme-to-substrate ratio and digested at 37 °C for 16–18 h.

### TMT labeling of peptides and HPLC fractionation

Desalted peptides were labelled using TMT6/10-plex reagents (TMT6/10 plex Isobaric Label Reagent Set, Thermo Fisher Scientific, Waltham, MA, USA) by the instructions. The labelling reagents were dissolved in acetonitrile. The differently labelled peptides were homogeneously mixed and incubated for 2 h and then desalted using a peptide desalting spin column (Thermo Fisher Scientific, Waltham, MA, USA).

TMT-labelled peptide mix was fractionated using a C18 column (Waters BEH C18 300 μm × 150 mm, 1.7 μm) on a Rigol L3000 HPLC. The operation was as follows: mobile phases A (20 mM ammonium formate, adjusted pH to 10.0) and B (100% acetonitrile) was used to develop a gradient elution. The solvent gradient was 3%–41% acetonitrile. And 20 components were collected after separation, which were concentrated into peptide powder by vacuum and stored at −20 °C.

### LC-MS/MS analysis

The present research employed EASY-nLC 1200 system (Thermo Fisher Scientific, Waltham, MA, USA) and Orbitrap Q Exactive HF-X mass spectrometer (Thermo Fisher Scientific, Waltham, MA, USA) to analyze the proteomics. The peptides were separated on a homemade analytical column with a linear gradient from 5% to 100% of eluent B (0.1% FA) using TMT-6 plex at a flow rate of 300 NL/min for eluent A (0.1% FA in water). The gradient of solvent was as follows: 3–5 percent of B, 5 s; 5–15 percent of B, 23 mins 55 s; 15–28 percent of B, 21 mins; 28–38 percent of B, 7 mins 30 s; 38-100 percent of B, 5 s; 100 percent of B, 12 mins 25 s.

### Data analysis

LC-MS/MS raw files were processed using Maxquant (v1.6.14) software for database search (Uniprot-sheep-proteome_UP000002356), building the database with the DDA method. Alkylation of cysteine was set as a fixed modification. Methionine oxidation and N-terminal acetylation of protein were set as a variable modification. The false discovery rate (FDR) of proteins and peptides was set at 0.01.

Different expression proteins (DEPs) were sifted by *P* value <0.05 and FC >1.2 or FC <0.83 [fold change, FC]. The function of DEPs was determined by gene ontology (GO) analysis through GoPlot (v1.0.2) and GO enrichment analysis of DEPs was performed from biological process (BP), molecular function (MF) and cellular component (CC). Then, the molecular process pathways for differential protein enrichment were obtained through the Kyoto Encyclopedia of Genes and Genomes (KEGG) database.

### Western blotting

Total proteins from C and P groups were lysed for 30 min using phenylmethanesulfonyl fluoride (PMSF) and separated with 10% sodium dodecyl sulfate-polyacrylamide gel electrophoresis (SDS-PAGE). After that, membranes were subjected to a standard blocking with 5% non-fat milk, hybridized with primary antibody (MAPK12, 1:500, 22665-1-AP; BCL2L1, 1:1000, R23529; UBA52, 1:500, 16432; MT-ND1, 1:500, A8035; GAPDH, 1:5000, 200306-7E4) at 4 °C overnight, and incubated with secondary antibody (goat anti-rabbit IgG, 1:500) at room temperature for 1 h. ImageJ software was used for grayscale analysis.

## Metabolomic analysis
### Metabolite extraction

First, 400 µl prechilled methanol-acetonitrile (3:1, v/v) used to be delivered to the GCs samples of C and P groups, and the samples were homogenized with the usage of a grinding mill after standing for 1 h. The extract was centrifuged at 3,000 × g for 15 min at 4 °C to obtain 400 µl supernatant, which was dried thoroughly. Next, the samples supplementation with 100 µl prechilled methanol-water (1:1, v/v), then the suspension was eddied (2,000 rmp, 4 °C, 3 min) and centrifuged (12,000 g, 4 °C, 10 min), to collect the supernatant for further analysis.

### LC-MS/MS analysis

LC-MS/MS analyses were conducted using Vanquish UHPLC system (Thermo Fisher Scientific, Waltham, MA, USA) with an Orbitrap Q Exactive HF-X mass spectrometer (Thermo Fisher Scientific, Waltham, MA, USA). Using a linear gradient, samples were injected into a ACQUITY UPLC HSS T3 (1.7 µm, 2.1 mm × 150 mm) column. The LC mobile phases A (5 mM ammonium formate), B (acetonitrile), C (0.1% formic acid) and D (0.1% formic acid acetonitrile) were used. The solvent gradient was set as follows: 2% B, 1 min; 2–50% B, 8 mins; 50–98% B, 3 mins; 98% B, 1.5 mins, 98–2% B, 1.5 mins and 2% B, 3 mins for the negative polarity mode. 2% D, 1 min; 2–50% D, 8 mins; 50–98% D, 3 mins; 98% D, 1.5 mins, 98–2% D, 1.5 mins and 2% D, 3 mins for the positive polarity mode. Q Exactive HF-X mass spectrometer was operated in positive/negative polarity mode with a spray voltage of 2.5 kV, capillary temperature of 325 °C, auxiliary temperature of 300 °C, sheath gas flow rate of 30 arb and aux gas flow rate of 10 arb.

### Data analysis

Peak alignment, peak picking and quantitative analysis of raw data were performed by Compound Discoverer 3.0 (CD 3.0, Thermo Fisher Scientific, Waltham, MA, USA). The main parameters were set as to the methods described previously (*He et al., 2021*).

In LC-MS metabolomic analysis, the following criteria were employed for selecting the differentially expressed metabolites (DEMs) in C and P groups: variable influence on projection (VIP) value >1.0, $P$ value <0.05 and |FC|>1.5 [fold change, FC]. The DEMs of both groups their enriched KEGG pathway were analyzed.

## Statistical analysis

Statistical analysis was performed using SPSS software (ver. 22.0, IBM Corp. Armonk, NY, USA). One-way ANOVA followed by Duncan's *post hoc* test, was used to compare the cell activity and oxidative stress related indicators among multiple groups. Statistical comparisons between C and P groups were performed using the $t$-test. $P < 0.05$ indicated significance.

Multivariate statistical analyses, principal component analysis (PCA), relative standard deviation (RSD), Pearson's correlation coefficient, supervised partial least squares discriminant analysis (PLS-DA) and orthogonal partial least squares discriminant analysis (OPLS-DA) were conducted to evaluate data quality control. For the integrated analysis of the proteome and metabolome, the KEGG pathway database was used to identify relevant metabolic processes associated with DEPs and DMs in both groups. Additionally, information regarding their interaction network was also obtained.

# RESULTS

## Cytotoxicity and cell viability

The morphology and viability of GCs are shown in Fig. 1. Most of the normal adherent GCs (Fig. 1A) were fusiform with good refraction under the microscope. The density of GCs increased with four ng/mL PRL (Fig. 1B), indicating that four ng/mL PRL promoted GCs proliferation. The density of GCs did not decrease significantly with 20 ng/mL PRL (Fig. 1C), indicating that the cytotoxicity was low when the concentration of PRL was 20 ng/mL. The normal spindle-shaped and dead spherical of GCs coexisted when the concentration of PRL was 100 (Fig. 1D) or 500 (Fig. 1E) ng/mL. The number and the density of GCs decreased, and the proportion of GCs transformed from spindle to spherical increased along with the increase of the concentration of PRL, indicating that the toxicity increased with the increase of PRL concentration.

The CCK-8 assay results (Fig. 1F) showed that PRL increased cell viability significantly at the dose of four ng/mL ($P = 0.024$), while 20 ($P = 0.013$), 100 ($P < 0.001$) and 500 ng/mL ($P < 0.001$) PRL obviously inhibited cell viability.

## Oxidative stress parameters analysis

The contents of MDA, SOD and T-AOC were shown in Figs. 2A–2C. Compared with the 500 ng/mL group, the content of MDA was repressed in the 0 ($P = 0.006$), 4 ($P = 0.004$), 20 ($P = 0.006$) and 100 ($P = 0.0255$) ng/mL groups, while the content of MDA was higher in the 100 ng/mL group than in the 0 ($P = 0.018$), 4 ($P = 0.006$) and 20 ($P = 0.020$) ng/mL groups, and the content of MDA was elevated in the 0 ($P < 0.001$) and 20 ($P < 0.001$) ng/mL groups compared with the four ng/mL group. The activity of SOD was increased in the 0 ($P = 0.004$), 4 ($P < 0.001$), 20 ($P = 0.004$) and 100 ($P = 0.044$) ng/mL groups

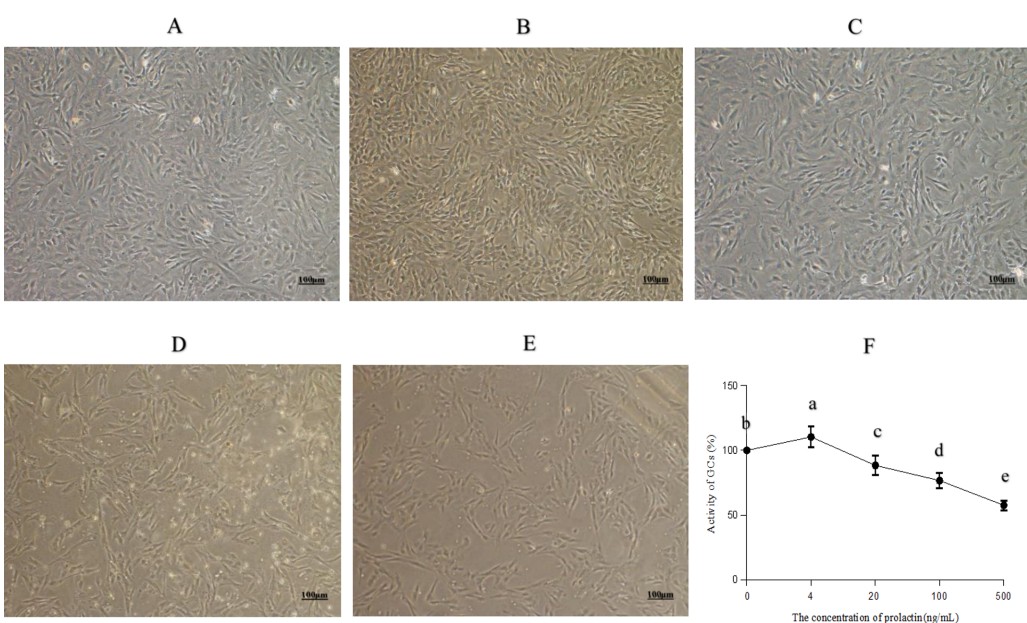

**Figure 1 Morphology and activity of ovine granulosa cells (GCs).** (A) Morphology of normal GCs (0 ng/mL PRL), (B) morphology of GCs with four ng/mL PRL, (C) morphology of GCs with 20 ng/mL PRL, (D) morphology of GCs with 100 ng/mL PRL, (E) morphology of GCs with 500 ng/mL PRL, and (F) cell activity of granulosa cells. The different lowercase letters indicate significant differences ($P < 0.05$). PRL, Prolactin.

compared with the 500 ng/mL group, while the activity of SOD was lower in the 100 ng/mL group than in the 0 ($P = 0.012$), 4 ($P < 0.001$) and 20 ($P = 0.011$) ng/mL groups, and the activity of SOD was increased in the 0($P = 0.002$) and 20 ($P < 0.001$) ng/mL groups compared with the four ng/mL group. Relative to the 500 ng/mL group, the T-AOC was decreased in the 0 ($P = 0.003$), 4 ($P = 0.002$), 20 ($P = 0.015$) and 100 ($P = 0.017$) ng/mL groups, while the T-AOC was lower in the 100 ng/mL group than in the 0 ($P = 0.005$) and 4 ($P = 0.003$) ng/mL groups. The level of ROS was shown in Fig. 2. Intracellular ROS was evaluated by fluorescein isothiocyanate-A(FITC-A). With the increase of PRL concentration, the level of ROS decreased first and then increased. The forward scatter area and FITC-A of ROS were shown in Figs. 2D–2E. Relative to the 500 ng/mL group, the level of ROS was decreased in the 0 ($P < 0.001$), 4 ($P < 0.001$), 20 ($P < 0.001$) and 100 ($P = 0.007$) ng/mL groups, while the ROS was higher in the 100 ng/mL group than in the 0 ($P = 0.002$), 4 ($P = 0.001$) and 20 ($P = 0.002$) ng/mL groups, and the level of ROS was increased in the 0 ($P = 0.021$) and 20 ($P = 0.032$) ng/mL groups compared with the 4 ng/mL group.

## Proteomic analysis
### Protein identification and the analysis of DEPs
There were 821,838 LC-MS/MS spectra matched to the known spectra, whereas the number of available spectra was 120,505. Overall, 7,552 credible proteins and 85,761 peptides from all GCs samples were detected and quantified by 1% FDR. However, only 7,499 proteins

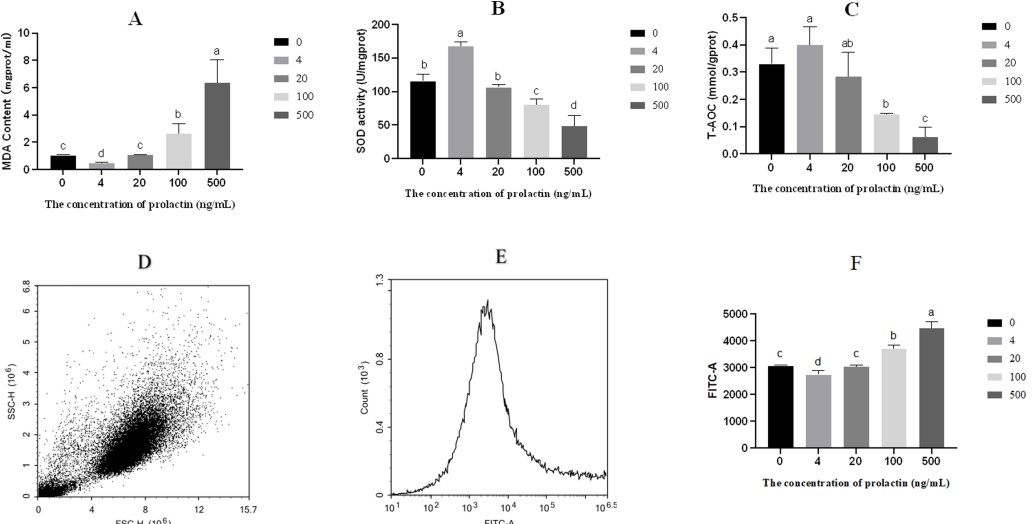

**Figure 2  Oxidative stress parameters.** (A) The content of MDA, (B) the activity of SOD, (C) T-AOC, (D) forward scatter area of ROS, (E) FITC-A of ROS, (F) the content of ROS. The different lowercase letters indicate significant differences ($P < 0.05$). MDA: Malondialdehyde; SOD: Superoxide Dismutase; T-AOC: Total antioxidant capacity; ROS, Reactive oxygen species.

could be quantified (Fig. 3A). PCA, RSD and Pearson's Correlation Coefficient were performed to evaluate the reproducibility of protein quantification. The PCA plot (Fig. 3B) indicated that the proteomic analysis was reliable; RSD (Fig. 3C) showed that the whole samples were more repetition, and Pearson's correlation coefficient (Fig. 3D) indicated the samples had a high degree of similarity.

According to the criteria of *P* value <0.05 and FC >1.2 or FC <0.83, Relative to the C group, 93 proteins among all credible proteins were significantly downregulated or upregulated in P group. The volcano map and heatmap of the DEPs were shown in Figs. 3E and 3F, respectively. Among the 93 DEPs, P0C276 (UBA52), Q9MZS7 (BCL2L1) and W5QEW6 (MAPK12) were associated with mitophagy. O78747 (MT-ND1) was contacted with reactive oxygen species (ROS). Q9MZS7 (BCL2L1) and W5QEW6 (MAPK12) enriched with functions correlated with apoptosis and NOD-like receptor signaling pathway. P0C276 (UBA52) and W5PQR5 (RPL18) were involved in ribosome. Q9MZS7 (BCL2L1) and W5QEW6 (MAPK12) were closely related to p53 signaling pathway, and 078747 (MT-ND1) in connection with oxidative phosphorylation (Table 2). We found that the DEPs involved in antioxidant related functions were significantly upregulated in the P group. Here, four proteins (UBA52, BCL2L1, MT-ND1 and MAPK12) related to OS were selected for validation (Fig. 4). The results of western blotting verified that these protein expression levels in P group were significantly higher than C group, which consistent with the proteomic findings.

### GO functional classification and KEGG pathway analysis of DEPs

In order to obtain in-depth understanding of the biological significance, the enrichment of DEPs according to GO were tested. As a result, 102 GO terms significantly enriched for

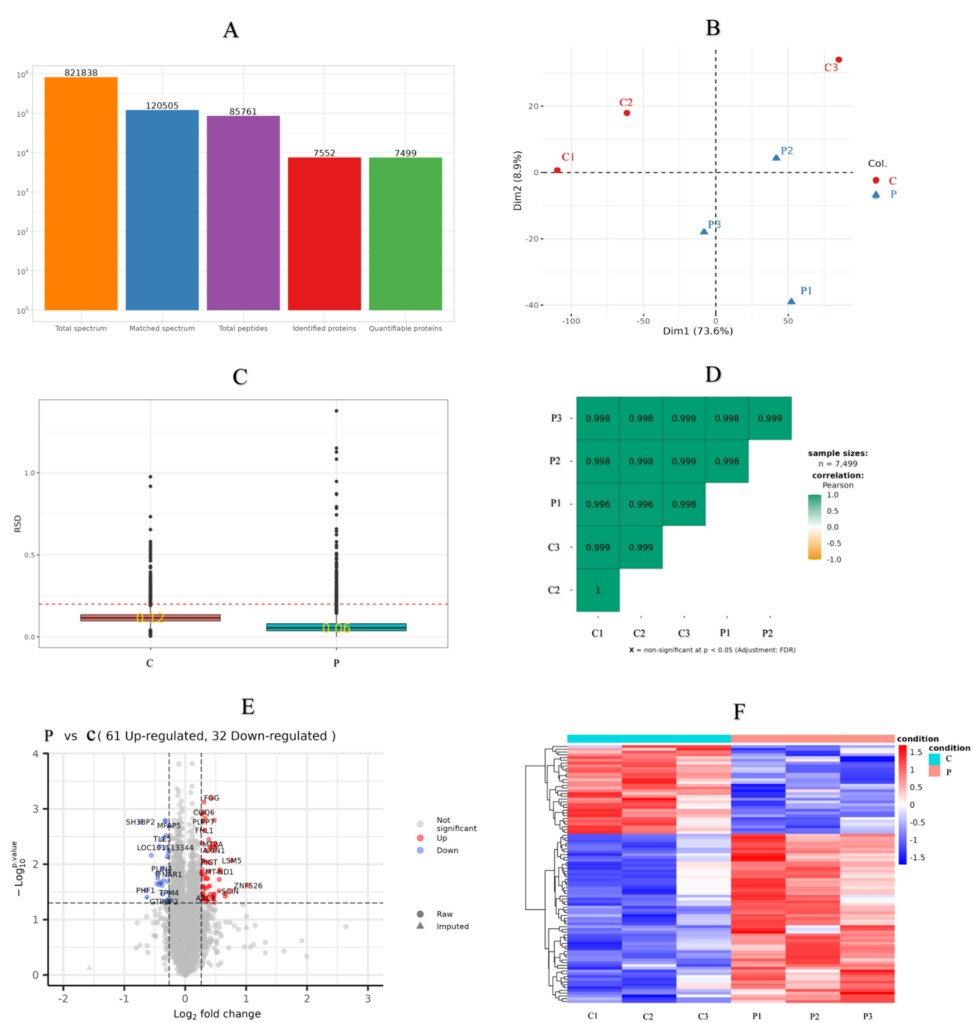

**Figure 3** **Sample repeatability test and DEPs in the proteomic.** (A) An overview of protein identification, (B) PCA score plot exhibiting significant difference between C (control group: 0 ng/mL PRL) and P (500 ng/mL PRL) groups ($n = 6$), (C) relative standard deviation, (D) Pearson's correlation coefficient, (E) volcano plot depicting the DEPs in C and P groups ($n = 6$), and (F) hierarchical clustering of the DEPs between the two groups ($n = 6$). DEPs, differentially expressed proteins. A–F, C: control group (0 ng/mL PRL); P: P group (500 ng/mL PRL).

**Table 2** **Differentially expressed proteins induced by high PRL.**

| Genes | Proteins | Gene ID | P-value |
|-------|----------|---------|---------|
| UBA52 | P0C276 | 443296 | 0.035 |
| MT-ND1 | O78747 | 808249 | 0.014 |
| BCL2L1 | Q9MZS7 | 443061 | 0.035 |
| MAPK12 | W5QEW6 | 101115956 | 0.048 |
| RPL18 | W5PQR5 | 101122808 | 0.004 |

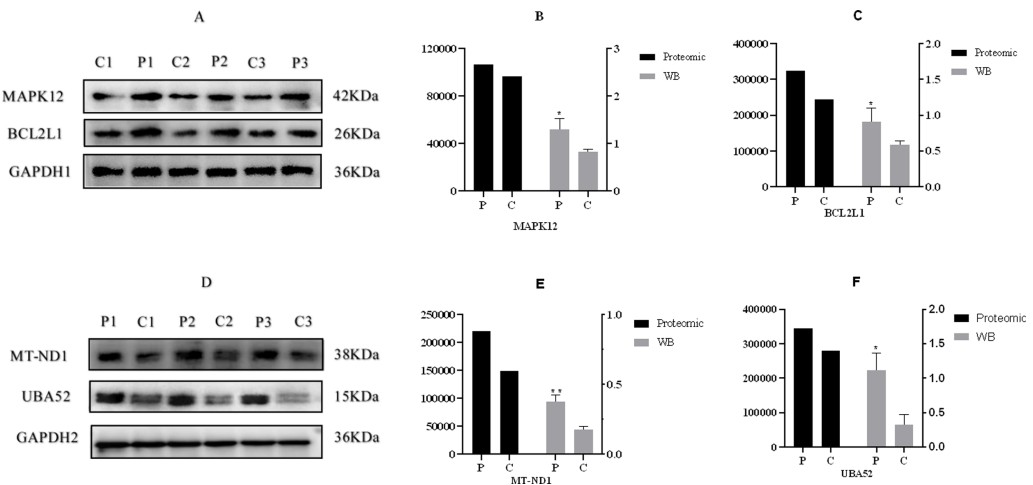

**Figure 4** **Western blotting validation of proteomic.** (A) & (D), Western blotting result, (B), (C), (E) & (F), the expression of four proteins. "*" and "**" indicates $0.01 < P < 0.05$ and $P < 0.01$, respectively . A–F, C: control group (0 ng/mL PRL); P: P group (500 ng/mL PRL). The Western blotting data is referenced on the right $Y$-axis, while the proteome data is referenced on the left $Y$-axis.

DEPs consisted of BP, CC and MF for which the number was 56, 24 and 22, respectively. Several GO terms significantly enriched for DEPs were found and shown in Fig. 5A. Among these pathways, the DEPs were mainly associated with the following BPs: mitochondrial electron transport, NADH to ubiquinone, ATP synthesis coupled electron transport, oxidative phosphorylation, electron transport chain, cellular respiration, regulation of ATPase activity, and RNA catabolic process. The main functional groups of DEPs in CC were nuclear membrane, ribosome and actomyosin, and in MF were actin binding, complement binding and NADH dehydrogenase activity.

Significantly enriched pathways in DEPs were analyzed by the KEGG database. The results revealed that these differently express target proteins were able to be mapped to 53 signaling pathways. Among these signaling pathways, the top 20 regulated pathways ranked based on the $P$-value were presented in Fig. 5B. The corresponding DEPs were mainly associated with pathways related to the following: mitophagy (animal), oxidative phosphorylation, apoptosis, NF-kappa B signaling pathway, ribosome, p53 signaling pathway, oxidative phosphorylation, NOD-like receptor signaling pathway, and chemical carcinogenesis—reactive oxygen species.

### Protein-protein interaction analysis
The STRING database was used to further develop the protein-protein interaction (PPI) networks for DEPs. These interactions include both indirect functional links and direct physical connections.

In Fig. 5C, down-regulated proteins are represented by blue nodes, while up-regulated proteins are represented by red nodes. The P0C276 and W5PQR5 proteins greatly influenced the regulation of ribosome, and the findings revealed that some proteins could not interact with one another directly. However, they still play a role through HPC.

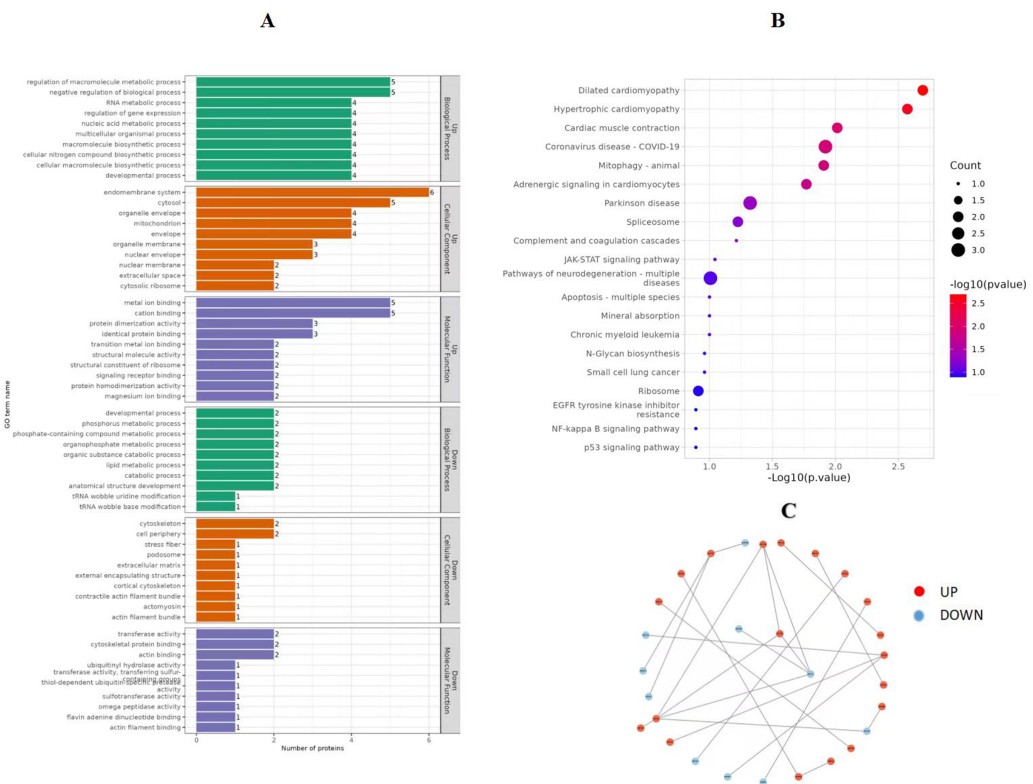

**Figure 5 GO function enrichment and KEGG pathways of DEPs (P vs C).** (A) GO function enrichment of DEPs, (B) KEGG pathways of DEPs, and (C) protein-protein interaction analysis. GO, Gene Ontology. KEGG, Kyoto Encyclopedia of Genes and Genomes. DEPs, differentially expressed protein. C, control group (0 ng/mL PRL). P, P group (500 ng/mL PRL).

## Metabolomic analysis
### DEMs identified

The difference in specific metabolites is likely due to the different concentrations of PRL. The metabolites were identified following the treatments with 0 and 500 ng/mL PRL by LC-MS/ MS. A total of 1,118 metabolites were identified using the positive ion (POS) mode, and 809 were identified using the negative ion (NEG) mode (Table S1). The OPLS-DA (Figs. 6A and 6B) and PLS-DA (Figs. 6C and 6D) were carried out in two groups to assess the quality control of metabolites. These data suggested that the experimental data were well controlled.

We compared the identified metabolites of ovine ovarian granulosa cells (GCs) from HPC and C groups to elucidate the underlying regulatory mechanism. The results were shown in Fig. 7. In the POS mode (Fig. 7A), 41 DEMs were up-regulated and 15 down-regulated; in the NEG mode (Fig. 7B), 40 DEMs were up-regulated and 38 down-regulated. These DEMs were mainly organic acids, lipids, inorganic alkanes, alcohols, carbohydrates, aldehydes, and ketones at POS and NEG modes. The relative abundance of these DMs was shown in Fig. 7C (POS) and Fig. 7D (NEG).

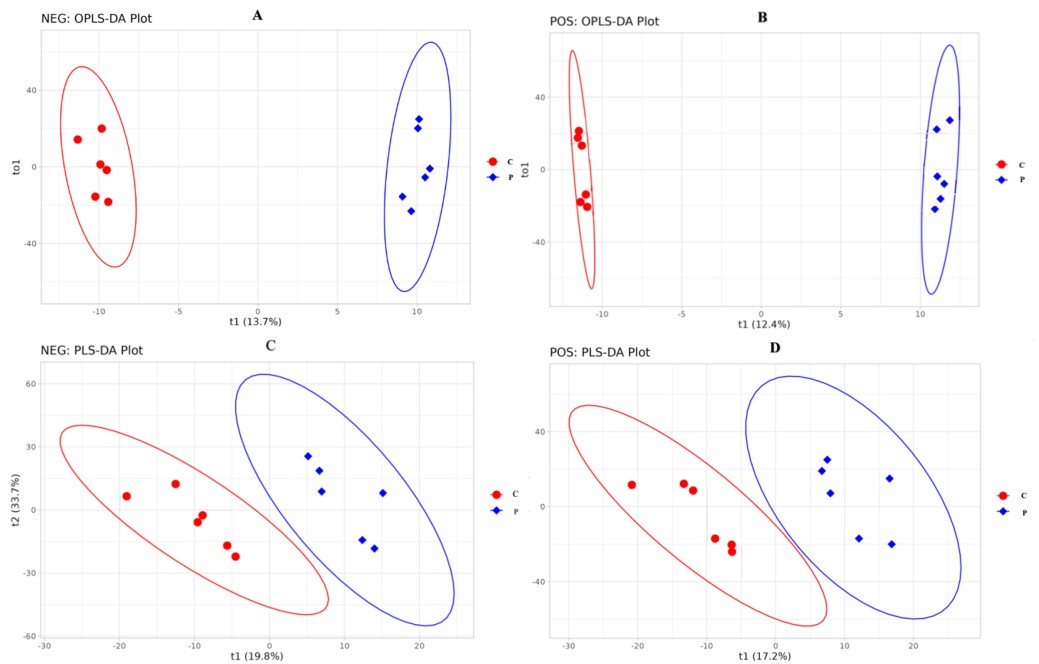

**Figure 6** **OPLS-DA and PLS-DA score plot of C and P groups (P *vs* C).** NEG, negative ion. POS, positive ion. (A–D), C: Control group (0 ng/mL PRL); P: P group (500 ng/mL PRL).

**Table 3** **Differentially expressed metabolites induced by high PRL.**

| ID | Name | Formula | *P*-value |
|---|---|---|---|
| POS0391 | (R)-Lactate | C3H6O3 | 0.010 |
| POS0646 | 2-Carboxy-D-arabinitol1-phosphate | C6H13O10P | 0.028 |
| NEG0427 | D-Erythrose4-phosphate | C4H907P | 0.036 |
| NEG0085 | 6-Phosphogluconic acid | C6H13O10P | 0.028 |

### KEGG pathway analysis of DEMs

The pathway analysis utilised the KEGG database to identify the significantly enriched biochemical processes in DEMs. The results showed that the signaling pathways included 14 pathways at POS mode and 11 pathways at NEG mode. After pathway enrichment analysis, the regulated pathways ranked based on the *P*-value were presented in Fig. 7. The corresponding DEMs at POS mode (Fig. 7E) were mainly associated with pathways related to the following: tryptophan metabolism, cAMP signaling pathway, the pentose phosphate pathway, and purine metabolism. In the NEG mode (Fig. 7F), the DEMs were involved in several KEGG pathways, including purine metabolism, pentose phosphate pathway, and pyrimidine metabolism. Among them, the DEMs enriched in the pentose phosphate pathway were as follows (Table 3): (R)-Lactate and 2-Carboxy-D-arabinitol1-phosphate in POS mode, and D-Erythrose4-phosphate and 6-Phosphogluconic acid in NEG mode.

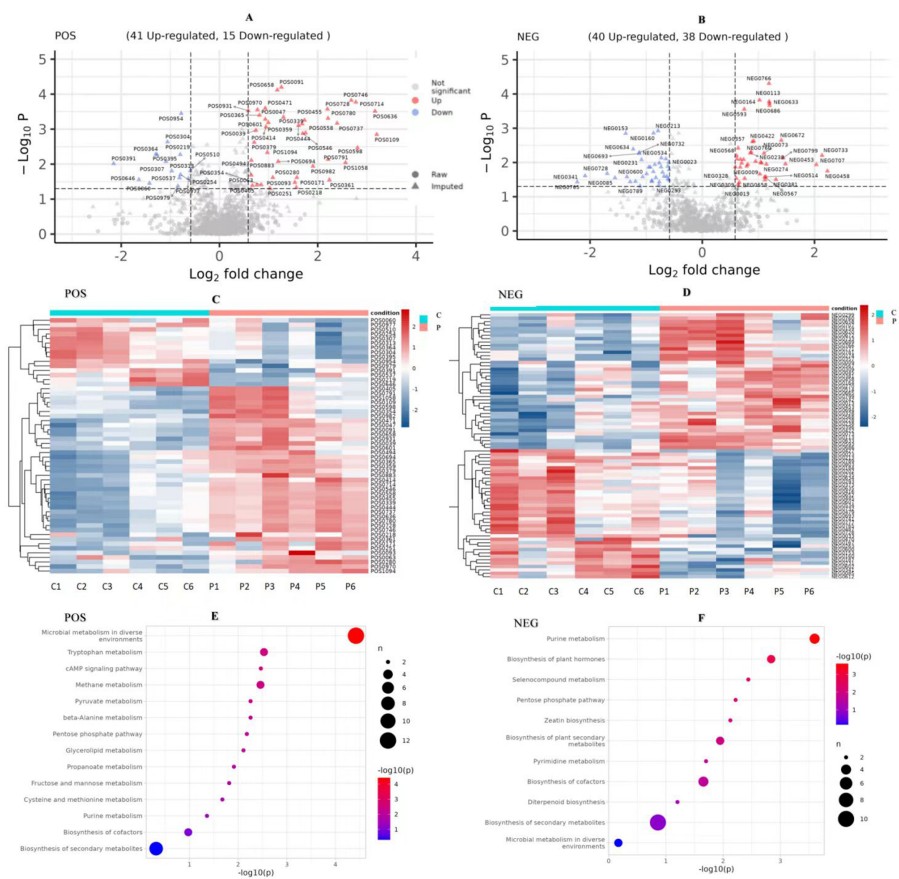

**Figure 7** **Analysis of metabolomic between C and P groups (P *vs* C).** (A) Volcano map of DEMs at POS mode, (B) volcano map of DEMs at NEG mode, (C) cluster heat map of DEMs at POS mode, (D) cluster heat map of DEMs at NEG mode, (E) KEGG pathways of DEMs at POS mode, and (F) KEGG pathways of DEMs at NEG mode. DEMs, differentially expressed metabolites. POS, positive ion MODE. NEG, negative ion mode. KEGG, Kyoto Encyclopedia of Genes and Genomes. C, control group (0 ng/mL PRL). P, P group (500 ng/mL PRL).

### Integrated omics analysis of proteomics and metabolomics

An integrated analysis based on the aforesaid omics data was conducted to identify the related enriched pathways of the DEPs and DEMs. As a result, Therefore, the pathway information was captured by mapping these DEPs to the KEGG pathway database (Fig. 8). Overall, four enriched KEGG pathways related to OS based on DEPs were obtained, which mainly included mitophagy (animal), apoptosis, ribosome, and chemical carcinogenesis—reactive oxygen species. The metabolomic results showed that the DEMs in the POS and NEG modes were both enriched in the pentose phosphate pathway, and NADPH produced in the pentose phosphate pathway was critical for alleviating ROS-related cell damage.

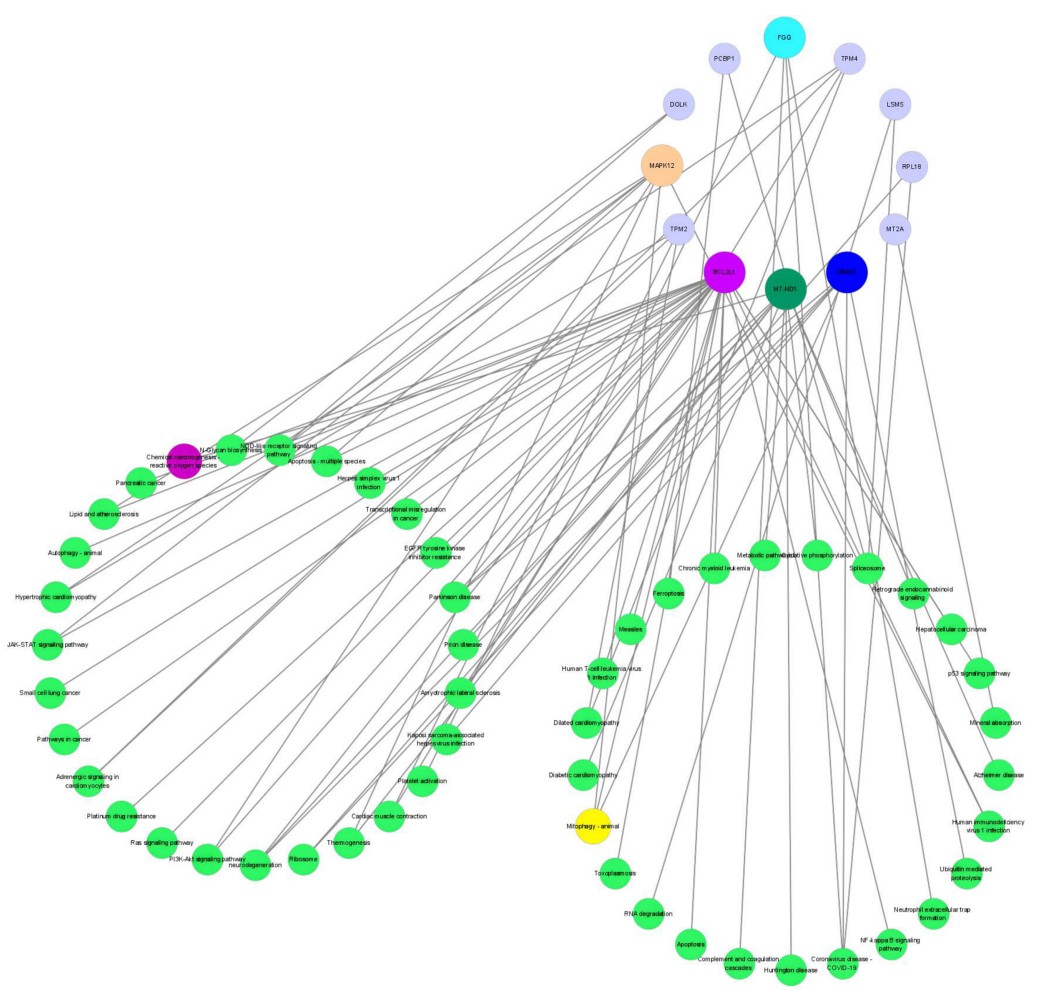

**Figure 8  The networks regulation of DEPs.** DEPs, Differentially expressed proteins.

## DISCUSSION

### The effect of PRL on cytotoxicity and oxidative stress in ovine ovarian GCs

PRL has been reported to be associated with cell proliferation and apoptosis, with low concentration of PRL promoting cell proliferation and high concentration leading to apoptosis (*Zhang et al., 2016a*; *Zhang et al., 2016b*). In our study, the low concentration (four ng/mL) of PRL promoted cell proliferation and inhibited cytotoxicity, while HPC showed the opposite effect in ovine ovarian GCs, which was consistent with the previous study. Additionally, PRL has also been reported to play an important role in oxidative-antioxidant balance (*Farmer, Lapointe & Cormier, 2017*; *Rodak et al., 2022*). Previous study showed that a moderate dose of PRL promotes the antioxidant capacity of adult RPE 19 human cells by reducing glutathione (*Melendez Garcia et al., 2016*), suggesting PRL could exert an antioxidant action. However, several studies have discovered that HPC leads to oxidative stress in mammals (*Farmer, Lapointe & Cormier, 2017*), with an increase in ROS

generation and alterations to antioxidant system components (*Rodak et al., 2022*), which is consistent with the results of the present study. In our study, the level of ROS in the four ng/mL group was decreased, the content of MDA depressed, and the levels of SOD and T-AOC were increased compared to other groups, but the results of the 500 ng/ml group showed the opposite trend. Previous studies suggested that pathologic conditions produced high ROS levels in ovarian follicles, leading to oxidative stress and extensive GCs damage (*Shen et al., 2016*; *Esfandyari et al., 2021*). Therefore, a possible explanation for the ROS increase in the present trial might be due to PRL-induced cytotoxicity in the GCs. However, the detailed mechanism that HPC causes oxidative stress in GCs stills needs further investigation.

## Integrated omics analysis of proteomic and metabolomic

Mitochondria are a unique dynamic double membrane-bound organelle (*Hagberg et al., 2014*) that perform an essential regulatory function in apoptosis (*Abate et al., 2020*) and ROS production (*Protasoni & Zeviani, 2021*). When electron transport in the mitochondrial respiratory chain is impaired and decoupled from ATP, which leads to the production of ROS (*Sommer et al., 2016*). ROS generated by oxidative stress can seriously affect cell development, function and survival while possibly inflicting damage to intracellular macromolecules and mitochondria through various signalling pathways. Mitochondria are important in deciding oocyte developmental competence. GCs around oocytes, on the other hand, can enhance mitochondrial function *via* mitophagy, thereby enhancing oocyte developmental ability (*Zhang et al., 2022*). Previous studies have shown that the mitochondria from human GCs are involved in oocyte maturation and embryo development (*Boucret et al., 2015*; *Ogino et al., 2016*). Moreover, melatonin (*Jiang et al., 2021*) and FSH (*Besnard, Horne & Whitehead, 2001*) repressed mitophagy to protect mouse GCs from oxidative damage. *Olavarria, Figueroa & Mulero (2012)* reported that physiological concentrations of native PRL were able to induce the production of ROS in head kidney leukocytes and macrophages from the teleost fish gilthead seabream and PRL is among the hormones that can impact mitochondrial function and modulate the underlying adaptations to changing bioenergetic and metabolic needs (*Alvarez-Delgado, 2022*), suggesting that PRL can act on oxidative stress in ovine ovarian GCs through ROS pathway and mitophagy pathway. It is consistent with the results of this study. In this study, we identified the cellular responses in ovine ovarian GCs by analyzing proteomes after handling with 500 ng/mL PRL. The results showed that the ROS pathway and the mitophagy pathway were changed significantly.

In the present study, MT-ND1 enriched in the ROS pathway, UBA52, BCL2L1 and MAPK12 enriched in the mitophagy pathway were screened by proteomics. The study showed that higher expression of MT-ND1 in the recurrent implantation failure (RIF) group compared with the healthy group may be related to increased oxidative stress in the endometrium (*Eker et al., 2021*). High mitochondrial oxidative phosphorylation (mt-OXPHOS) levels might be generated excessive ROS and have an adverse effect on follicular health (*Hoque et al., 2021*). Ubiquitin-52 amino acid fusion protein (UbA52) is translated and expressed by UbA52 gene which participates in oxidative stress, ribosome and the

ubiquitin-proteasome pathway (UPP) (*Yang & Zhang, 2014*). Under hypoxia conditions, ubiquitin protein UBA52 was down-regulated and ubiquitinated through its interaction with apoptosis-inducing factors, resulting in mitosis and abnormal autophagy (*Ma et al., 2022*). MAPK12 is a key member of the P38 (MAPK) pathway (*Cuadrado & Nebreda, 2010*). It has been reported that physalin A can induce ROS-mediated apoptosis, and autophagy plays a protective role through the p38 (MAPK)-NFKB/NF-kappaB survival pathway (*He et al., 2013*). There was also a study which showed that copper-induced oxidative stress will induce protective autophagy through transcriptional regulation of autophagy genes by activation of the MAPK pathway in HeLa cells (*Zhong et al., 2014*). BCL2L1, as a member of Bcl-2 family, was up-regulated under oxidative stress (*Wang et al., 2019*) and the expression of *BCL2L1* mRNA levels, encoding for *BCL-xL*, was down-regulated through oxidative stress by dehydration in cortex (*Ali et al., 2020*). Bcl-2 can induce the necrotic type of death in human coronary artery endothelial cells through oxidative stress (*Maslanakova et al., 2016*). A study showed that the human adipose-derived mesenchymal stem cells exposure to GC-DNA increased oxidative stress along with the increase of BCL2L1 (*Kostyuk et al., 2015*). Therefore, differential proteins MT-ND1 enriched in the ROS pathway and UBA52, BCL2L1 as well as MAPK12 enriched in the mitophagy pathway were closely related to oxidative stress.

PRL binding to the prolactin receptor (*PRLR*) exerts pleiotropic biological effects in mammals (*Hu, Zhuang & Dufau, 1998*). *Nie et al. (2021)* showed that short *PRLR* regulates the pentose phosphate pathway in human by reducing the expression of two rate-limiting enzymes in pentose phosphate metabolism, G6PD and TKT. In this study, we screened the pentose phosphate pathway by both POS and NEG modes of the metabolomic. Researchers explored that HK2 attenuates cardiac hypertrophy by decreasing ROS accumulation *via* increased pentose phosphate pathway flux (*McCommis et al., 2013*). *Agarwal et al. (2021)* showed that Parkin, a key effector of mitophagy altered in Parkinson's disease, inhibits the pentose phosphate pathway, which creates metabolic and oxidative stress. BCL2L1, an important apoptosis regulating protein, is localized to the outer mitochondrial membrane (*Sillars-Hardebol et al., 2012*). Data also demonstrated that mitochondrial BCL2L1 is participate in keeping mitochondrial respiratory capacity, and it accumulated under oxidative stress (*Stiegler et al., 2013*), which is related with glycolytic capacity and balanced by increased pentose phosphate pathway activity (*Pfeiffer et al., 2017*). Hence, the pentose phosphate pathway can regulate oxidative stress through ROS and mitophagy, which is consistent with our experimental results. In this study, the pentose phosphate pathway was changed significantly in ovine ovarian GCs by analyzing metabolomic after handling with 500 ng/mL PRL.

## CONCLUSIONS

1. Low concentration of PRL inhibited cytotoxicity and oxidative stress while high PRL concentration induced cytotoxicity and oxidative stress.
2. High PRL concentration promoted oxidative stress in ovine ovarian GCs by the pentose phosphate pathway through regulating the related proteins MT-ND1 in the ROS pathway, and UBA52, MAPK12 and BCL2L1 in the mitophagy pathway.

## ACKNOWLEDGEMENTS

The authors express their gratitude to fellow students in the laboratory for assistance.

### Funding

This research was supported by the China Agriculture Research System (CARS-38) and (CARS-39-23); the Innovative ability training funding project for graduate students supported by the Hebei Provincial Department of Education (CXZZBS2022049). The funders had no role in study design, data collection and analysis, decision to publish, or preparation of the manuscript.

### Grant Disclosures

The following grant information was disclosed by the authors:
China Agriculture Research System: CARS-38, CARS-39-23.
Hebei Provincial Department of Education: CXZZBS2022049.

### Competing Interests

The authors declare there are no competing interests.

### Author Contributions

- Ruochen Yang conceived and designed the experiments, performed the experiments, analyzed the data, prepared figures and/or tables, authored or reviewed drafts of the article, and approved the final draft.
- Shuo Zhang performed the experiments, analyzed the data, authored or reviewed drafts of the article, and approved the final draft.
- Chunhui Duan performed the experiments, analyzed the data, authored or reviewed drafts of the article, and approved the final draft.
- Yunxia Guo analyzed the data, prepared figures and/or tables, and approved the final draft.
- Xinyu Shan analyzed the data, prepared figures and/or tables, and approved the final draft.
- Xinyan Zhang analyzed the data, prepared figures and/or tables, and approved the final draft.
- Sicong Yue analyzed the data, prepared figures and/or tables, and approved the final draft.
- Yingjie Zhang conceived and designed the experiments, analyzed the data, authored or reviewed drafts of the article, and approved the final draft.
- Yueqin Liu analyzed the data, authored or reviewed drafts of the article, and approved the final draft.

### Data Availability

The mass spectrometry proteomics data are available at the ProteomeXchange Consortium via the iProX partner repository: PXD037645.

The raw data are available in the Supplemental Files.

## Supplemental Information

Supplemental information for this article can be found online at http://dx.doi.org/10.7717/peerj.15629#supplemental-information.

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
