# Peer review of "Effect of prolactin on cytotoxicity and oxidative stress in ovine ovarian granulosa cells"

_PeerJ, doi:10.7717/peerj.15629_

## Round 0.1 · original submission · Major Revisions

Dear authors,
thank you for your submission. Please, refer to the reviewers' comments for further details. Please, make sure to proofread the whole document, including the data presentation and statistical analysis.

·

Basic reporting

Good work
Good set of data
Fair presentation of data and crisp discussion.

However, Please address the following issues:

1. Details regarding GCs collection need to be explained: How you differentiate RBC and theca cells
2. Other criteria of cytotoxicity should be used other than morphology (Like MTT assays and other biochemical tests)
3. How GCs can be hatched... page 8, line no 121
4. Many sentences sytarts with "To.....". Please do not start any sentence with preposition
5. What is the normal PRL level in follicular fluid and serum in sheep and humans in normal conditions, pregnancy and different phases of cycle (Follicular and Luteal)?
6. Why you have chosen sheep as model?
7. Whether this study have any impact in domestic animal physiology or human or both... please explain.
8. Whether this study have impact in vitro cell culture study or in vivo conditions?
9. If in vivo, why 500 ng/ml of PRL? Is not this too high? In which conditions PRL level reach this much high in human and domestic animals... please explain
10. Grammatical errors and syntax errors should be checked.

Experimental design

1. Details regarding GCs collection need to be explained: How you differentiate RBC and theca cells
2. Other criteria of cytotoxicity should be used other than morphology (Like MTT assays and other biochemical tests)
3. How GCs can be hatched... page 8, line no 121
4. Many sentences sytarts with "To.....". Please do not start any sentence with preposition
5. What is the normal PRL level in follicular fluid and serum in sheep and humans in normal conditions, pregnancy and different phases of cycle (Follicular and Luteal)?
6. Why you have chosen sheep as model?
7. Whether this study have any impact in domestic animal physiology or human or both... please explain.
8. Whether this study have impact in vitro cell culture study or in vivo conditions?
9. If in vivo, why 500 ng/ml of PRL? Is not this too high? In which conditions PRL level reach this much high in human and domestic animals... please explain

Validity of the findings

Please refer above

Additional comments

Nil

Reviewer 2 ·

Basic reporting

This paper explores the mechanism of the effects of PRL on cytotoxicity and oxidative stress in ovine ovarian granulosa cells, using different methods. It is well organized the English language requires minor changes. To make the Introduction more robust, please include the important role of adequate ROS levels in reproductive function, citing recent papers.

Experimental design

Methods are appropriate for the purpose of study.

Validity of the findings

The results are consistent, but it is suggested to reduce the discussion, maintaining, however, its most important content.

Additional comments

Abstract - In the methods section please mention that the granulosa cells are from ovine.
L. 75 – Please cite references at the end of the sentence: Additionally…
L. 78 - Similarly, much evidence has shown…
L.83 – Please remove etc.
For aims of study please mention that the granulosa cells are from ovine.
L. 114 – Please justify doses of prolactin used.
L. 116 – Please consider only (Normal ovarian GCs 117 are usually spindle-shaped).
L. 139 – Then instead of Afterward
L.260-272 - Concerning to oxidative stress parameters it is suggested to include a table with the values, to make their analysis easier.
L. 282, 283 – spectra
L. 347 – Please rephrase the sentence to avoid the repetition of the word “pathway”.
Please consider the following paper:
Lai Q, et al., Oxidative stress in granulosa cells contributes to poor oocyte quality and IVF-ET outcomes in women with polycystic ovary syndrome. Front Med. 2018, 12(5):518-524.

---

## Round 0.2 · Minor Revisions

Dear authors. Thank you for your submission. The manuscript still requires fine adjustments, namely by incorporating some of the answers in the rebuttal into the manuscript - or presenting reasons why you wish not to do so—many thanks.

·

Basic reporting

Widely improved version
However, incorporate all the responses of the comments you have done in the rebuttal (response) section in the text of the manuscript with references. It will improve the quality of the paper.

Experimental design

Good

Validity of the findings

incorporate all the responses of the comments you have done in the rebuttal (response) section in the text of the manuscript with references. It will improve the quality of the paper.

Additional comments

incorporate all the responses of the comments you have done in the rebuttal (response) section in the text of the manuscript with references. It will improve the quality of the paper.

---

## Round 0.3 · accepted · Accept

Dear authors, thank you for your submission. I believe that your manuscript is now in a condition to be ACCEPTED for publication. Congratulations!